# Discrete-Time Fractional, Variable-Order PID Controller for a Plant with Delay

**DOI:** 10.3390/e22070771

**Published:** 2020-07-14

**Authors:** Piotr Oziablo, Dorota Mozyrska, Małgorzata Wyrwas

**Affiliations:** Faculty of Computer Science, Bialystok University of Technology, 15-351 Bialystok, Poland; d.mozyrska@pb.edu.pl (D.M.); m.wyrwas@pb.edu.pl (M.W.)

**Keywords:** fractional differences, PID control, variable-order

## Abstract

In this paper, we discuss the implementation and tuning algorithms of a variable-, fractional-order Proportional–Integral–Derivative (PID) controller based on Grünwald–Letnikov difference definition. All simulations are executed for the third-order plant with a delay. The results of a unit step response for all described implementations are presented in a graphical and tabular form. As the qualitative criteria, we use three different error values, which are the following: a summation of squared error (SSE), a summation of squared time weighted error (SSTE) and a summation of squared time-squared weighted error (SST2E). Besides three types of error values, obtained results are additionally evaluated on the basis of an overshoot and a rise time of the output signals achieved by systems with the designed controllers.

## 1. Introduction

Proportional–Integral–Derivative (PID) controllers are undoubtedly used in most automatic process control applications in the industry today. They are renowned for example as an excellent tool to regulate flow, temperature, pressure, and many other industrial process variables. Historically, the first PID controllers were built in 1939 by the Taylor and Foxboro instrument companies. Despite the various tuning methods most of present-day controllers are based on those original proportional, integral, and derivative modes. One of the first methods of PID tuning was Ziegler–Nichols, described in [1], which was widely extended along last decades. One of the directions of making PID controllers more robust is the use of fractional-order operators for both cases: continuous-, and discrete-time models. The review of fractional controllers is presented in [2]. The fractional PID controller was introduced by I. Podlubny in [3]. After this work, various authors designed fractional PID controllers using different tuning methods. It was shown in many papers, see for example [4,5,6,7,8,9], that the use of fractional PID controllers could make the entire control system perform better. The reason is that FOPID (fractional-order PID) controllers, in addition to standard $K_{p}$, $K_{i}$, $K_{d}$, introduce two more tuning parameters—integral and derivative order values, which provide wider flexibility during the control process. Although most of the controllers are devoted for analog systems, nowadays, the control of a system performance is mainly carried out with the aid of microcontrollers and microprocessors, see [10]. Many processes described by an analog system can be controlled by a digital system. The switching between analog and digital signals is obtained by the sampling of signals and zero-order hold part. The generalised transfer function of analog fractional PID controller depends on three gains: $K_{p}$, $K_{i}$, $K_{d}$; and two fractional values: λ and μ. The transfer function of such controller is described by the following formula: $G_{PID}\left(s\right)=K_{p}+\frac{K_{i}}{s^{\lambda }}+K_{d}s^{\mu }$, where λ is the order of integration, and μ is the order of differentiator. All the classical PID controllers are particular cases of the fractional-order controller, where both orders are equal to one. For discrete-time (digital) FOPID parameter, λ is an order of summation, while μ is an order of difference part of the controller.

In the paper, the implementation of fractional-, variable-order PID (FVOPID) controller is described. FVOPID controller is a generalisation of FOPID (fractional order PID). The controller has two (one for integral and one for derivative) order functions instead of having constant order values assigned to integral and derivative actions. It means that both orders are changed during the control process. An introduction to variable-, fractional-order differences can be found in [11,12,13], and an example algorithm of finding parameters of FVOPID controllers is presented in [14]. One can find more about potential applications in modelling in [15,16,17], where authors provide a survey of the recent relevant literature and findings in primary definitions, models, numerical methods, and their applications of variable-order fractional differential equations. We emphasise that in our paper we use the digital version of fractional controllers, and we can compare obtained results to those used for example in [18]. However, we use difference operator based on Grünwald–Letnikov difference definition; in the paper [18], it can be compared to the type A operator. We propose a new method of finding parameters and order values. The paper describes implementation and parameters for the tuning algorithm of digital FVOPID controller which is implemented using FVOGLD (fractional-, variable-order Grünwald–Letnikov difference). It is assumed that the designed controller has five different order values of integral and five different values of derivative order functions that depend on the ratio between the current error and the setpoint. Implemented controller parameters are set up to minimise three error criteria which are integral squared error (SSE), integral squared time weighted error (SSTE), and integral squared time-squared weighted error (SST2E).

## 2. Materials and Methods

The most important part in our investigations plays the definition of fractional, variable-order *h*-difference of Grünwald–Letnikov type. In some papers, e.g., [14,18,19], this operator is referred as the type A. In the definition, there is a sequence $a_{k}^{\nu_{l}}$ of coefficients used depending on a given variable-order function. It is important to notice that the sequence tends quite fast to zero and the smallest parts of it, for the considered step, are multiplied by the oldest trials. Then, in a certain sense, the sequence tries to forget older values but it does not vanish them completely. One can find this in some past papers, where based on this matter, authors truncate the whole sequence after a fixed amount of steps. Due to the property of forgetting the sequence of coefficients, it is also called the oblivion function, see [11,20].

**Definition** **1.**
*For k,l∈Z and ${\left(\nu_{l}\right)}_{l\in Z}$*the oblivion function* is defined as $a_{k}^{\nu_{l}}=0$ for all k<0 and l∈Z, $a_{0}^{\nu_{l}}=1$ and*
(1)$$a_{k}^{\nu_{l}}={\left(-1\right)}^{k}\frac{\nu_{l}\left(\nu_{l}-1\right)\cdots \left(\nu_{l}-k+1\right)}{k!}\;$$
*for k>0.*


Formula (Equation 1) in Definition 1 can be translated into the recurrence with respect to $k\in N_{0}$: (2)$$a_{0}^{\nu_{l}}=1\;,a_{k}^{\nu_{l}}=a_{k-1}^{\nu_{l}}\left(1-\frac{\nu_{l}+1}{k}\right),\;k\geq 1\;.$$

Let h>0 be a sample step and let us denote hZ:={…,−2h,−h,0,h,2h,…}.

**Definition** **2.**
*(FVOGLD) Let h>0 and x:hZ→R be a bounded function. The fractional-, variable-order Grünwald–Letnikov h-difference operator of a function x with step h>0 and an order function ν:Z→R is defined as a finite sum*
(3)$$\Delta_{h}^{\nu_{k}}x_{k}=\frac{1}{h^{\nu_{k}}}\sum_{i=0}^{k}a_{i}^{\nu_{k}}x_{k-i}=\frac{1}{h^{\nu_{k}}}\left[\begin{array}{cccc} 1 & a_{1}^{\nu_{k}} & \cdots & a_{k}^{\nu_{k}} \end{array}\right]\left[\begin{array}{c} x_{k}\\ x_{k-1}\\ \cdots \\ x_{1}\\ x_{0} \end{array}\right]\;,$$
*where $x_{k}:=x\left(kh\right)$.*


Definition 2 as a fractional-, variable-order summation that can be treated as discrete-time version of fractional-, variable-order integration (if the sample step *h* goes to zero). Observe that Definition 2 agrees with definition of type A, presented for example in [18,19], only if an order function has non-negative values and can be treated as an extension of difference operators. For constant-order function ν≡α, the operator given in Definition 2 agrees with the Grünwald–Letnikov fractional-order backward difference if α≥0, or becomes fractional summation for α<0. For constant-order functions, we refer the reader to [21,22].

### A Digital Fractional-, Variable-Order PID Controller

A digital fractional-, variable-order PID controller can be defined in a similar way as a first-order PID. The implemented FVOPID controller of type A is given by the Equation (Equation 4)
(4)$$y_{k}=K_{p}e_{k}+K_{i}\Delta_{h}^{\mu_{k}}e_{k}+K_{d}\Delta_{h}^{\nu_{k}}e_{k}\;.$$

In Equation (Equation 4), coefficients $K_{p}$, $K_{i}$, $K_{d}$ are proportional, summation, and difference gains respectively; $\mu_{k}$ and $\nu_{k}$ are values of order functions of summation and difference actions; *k* is the number of a sample; h>0 is a sampling time; $e_{k}$ is an input; and $y_{k}$ is an output of the controller. It is worth mentioning that if both order functions $\mu_{k}$ and $\nu_{k}$ are non-negative at the same time, then the controller acts like it has two differences with gains $K_{i}$ and $K_{d}$ and no summation. Similarly, if both order functions are simultaneously nonpositive, then the controller would work like it has two summations with gains $K_{i}$ and $K_{d}$ and no difference. One of the assumptions, which was taken during the controller implementation, was that there are five possible order values, $v_{i1},\ldots ,v_{i5}$ for summation; and five possible order values, $v_{d1},\ldots ,v_{d5}$ for difference operator. Selected number of order values is a compromise between control flexibility provided by variable-order controller (higher number of different order values could possibly give more control options) and computational time required by optimisation algorithms to find the order values (the time that optimisation algorithms require to find the optimal solution depends on the number of parameters). An additional assumption is that the order values depend on a ratio of an error value $e_{k}$ and a set-point value $u_{k}$, defined as
(5)$$\rho_{k}:=\frac{e_{k}}{u_{k}}\;.$$

Based on the current value of ratio $\rho_{k}$, the controller has summation and difference orders set according to the following rules:$v_{i1}$, $v_{d1}$ for the ratio $\rho_{k}>0.8$;$v_{i2}$, $v_{d2}$ for the ratio $\rho_{k}\in (0.6,0.8\rangle$;$v_{i3}$, $v_{d3}$ for the ratio $\rho_{k}\in (0.4,0.6\rangle$;$v_{i4}$, $v_{d4}$ for the ratio $\rho_{k}\in (0.2,0.4\rangle$;$v_{i5}$, $v_{d5}$ for the ratio $\rho_{k}⩽0.2$.
The main idea of designed variable-order controller is, in this case, to make the order values (which induces control characteristic) dependent on the current stage of the control process which is represented by the ratio (Equation 5). In order to find parameters of FVOPID controller, two versions of the tuning algorithm were developed. The first version of the algorithm (which will be referred as FOPID based) can be described by the following steps:Finding parameters $K_{p}$, $K_{i}$, and $K_{d}$ of the first-order PID controller using some selected method (e.g., Ziegler–Nichols or any other tuning approach).Using Nelder–Mead optimisation to find new parameters $K_{p}$, $K_{i}$, and $K_{d}$ of first-order PID controller (as the starting point of the optimisation, $K_{p}$, $K_{i}$, and $K_{d}$ values calculated in step 1 should be used).Using Nelder–Mead optimisation to find fractional order PID (FOPID) controller parameters (as the starting point of the optimisation, $K_{p}$, $K_{i}$, and $K_{d}$ values calculated in step 2 should be used). In this case, besides $K_{p}$, $K_{i}$, $K_{d}$ parameters, the optimisation algorithm additionally searches for optimal constant order values $v_{i}$ (summation) and $v_{d}$ (difference).Using Nelder–Mead optimisation to find new parameters $K_{p}$, $K_{i}$, and $K_{d}$ and order values $v_{i1}-v_{i5}$, $v_{d1}-v_{d5}$ of variable, fractional-order PID controller (as the starting point of the optimisation, $K_{p}$, $K_{i}$, $K_{d}$, $v_{i}$, $v_{d}$ values calculated in step 3 should be used).

The second version of the algorithm (which will be referred to as PID-based) is similar to the FOPID-based algorithm. The difference is that FVOPID parameters are searched starting from the parameters of optimal PID controller instead of FOPID controller. Note that by optimal PID controller we mean the PID controller designed with the Nelder–Mead optimisation to minimise given error criteria. Additionally, it was assumed that $v_{i5}$ and $v_{d5}$ order values should be set to 1 in this case. The reason of setting the order values to 1 (which means first order integral/derivative) for small values of error is to eliminate steady-state error which is characteristic for fractional order controllers. The PID-based algorithm can be described by the following steps:Finding parameters $K_{p}$, $K_{i}$, and $K_{d}$ of the first-order PID controller using some selected method (e.g., Ziegler–Nichols or any other tuning approach).Using Nelder–Mead optimisation to find new parameters $K_{p}$, $K_{i}$, and $K_{d}$ of first-order PID controller (as the starting point of the optimisation, $K_{p}$, $K_{i}$, and $K_{d}$ values calculated in step 1 should be used).Using Nelder–Mead optimisation to find new parameters $K_{p}$, $K_{i}$, and $K_{d}$ and order values $v_{i1}-v_{i4}$, $v_{d1}-v_{d4}$ of variable, fractional-order PID controller (as the starting point of the optimisation, $K_{p}$, $K_{i}$, $K_{d}$ values calculated in step 2 should be used).

Parameter searching procedures are based on minimisation of three types of summation errors which are as follows: sum squared error (SSE), sum squared time weighted error (SSTE) and sum squared time-squared weighted error (SST2E). Mentioned errors are calculated based on the Equations (Equation 6)–(Equation 8).
(6)$$SSE=\sum_{i=0}^{k}e^{2}\left(ih\right).$$
(7)$$SSTE=\sum_{i=0}^{k}{\left(ih\right)}^{2}e^{2}\left(ih\right).$$
(8)$$SST2E=\sum_{i=0}^{k}{\left(ih\right)}^{4}e^{2}\left(ih\right).$$

## 3. Results

The simulations were performed for the plant described by the following transfer function:(9)$$G_{s}=\frac{e^{-5s}}{{\left(s+1\right)}^{3}}\;.$$

The plant model was taken from [23], where the authors presented a design method for PID controllers based on constrained optimisation. The simulations were performed in Matlab Simulink with the sampling time set to h=0.02s. The simulation time used for controllers parameters search was set to 60 s. The initial parameters of PID controller (which corresponds to step 1 of the described algorithms) were set to $K_{p}=0.555$, $K_{i}=0.1729$, $K_{d}=0.9657$ according to [23]. The simulations were executed for both unconstrained and constrained control signal values. A rise time and an overshoot presented in the result tables were calculated with Matlab stepinfo function. A rise time in this case is considered as a time needed for the output signal to rise from 10% to 90% of the steady-state response [24].

### 3.1. Simulation Results for Controllers with Unconstrained Control Signal Values

During the first phase of the research, the simulations were performed with the assumption that there are no constraints set to the values of control signals generated by the controllers. As the starting point for the experiment, we chose a controller referred as Initial PID. By Initial PID, we understand PID controller with parameters set to $K_{p}=0.555$, $K_{i}=0.1729$, $K_{d}=0.9657$ (step 1 of previously described algorithms). The error values, rise time, overshoot, and minimum/maximum control values for the Initial PID controller setup are presented in Table 1. The step responses and the values of control signals generated by the controllers designed with SSE, SSTE, and SST2E error minimisation are presented in Figure 1, Figure 2, Figure 3, Figure 4, Figure 5, Figure 6, Figure 7, Figure 8 and Figure 9. To make figures more transparent/readable, presented simulation time was limited to 50 s (the parameters were searched for 60 s simulation). Additionally, control values presented in Figure 3 were limited to a range of −15 to 15 and in Figure 6 and Figure 9 to a range of −2 to 5. This is because at the beginning of the control process, the signals generated by the controllers reach very high values which would make analysis of the signals in later phase of the control difficult. The maximum/minimum controllers output signal values (which usually occur at the beginning of the simulation) can be found in the last two rows of Table 2, Table 3 and Table 4.

Optimal PID is a first-order controller, for which parameters were found by Nelder–Mead optimisation (steps 1 and 2 of described algorithms). FOPID controller is a fractional order PID for which parameters were found according to steps 1–3 of FOPID based algorithm. The values of implemented controllers parameters, rise time, overshoot, and minimised error of Optimal PID, FOPID, and FVOPIDs controllers are presented in Table 2, Table 3 and Table 4. Column ’Optimal PID‘ contains results for Optimal PID controller, ’FOPID‘ column represents results for Fractional order PID, while ’FVOPID-FO‘ and ’FVOPID‘ columns contain data respectively for FVOPID controller tuned with FOPID-based algorithm and FVOPID tuned with PID-based algorithm.

### 3.2. Simulation Results for Controllers with Constrained Control Signal Values

During the second phase of the research, the simulations were performed with the assumption that the absolute value of the control signal generated by the controllers should not exceed the maximum value returned by Initial PID (PID tuned with step 1). In this case, the controllers output signals are limited to the range of −48.8435 to 48.8435 (because the maximum value returned by the Initial PID controller is 48.8435). It is worth noticing that the limitation of the signal generated by the controllers is not achieved by adding saturation block to the controllers output, but by modifying the tuning algorithm so it searches for parameters for which the constraint criteria are met. This additional condition is simply achieved by checking the minimum and maximum controller output values in every optimisation step (for every set of parameters produced by Nelder–Mead optimisation) and if the values exceed the defined limit, then the maximum possible value is assigned to the minimised error (which forces the optimisation algorithm to search for another set of parameters). The obtained results for constrained controllers output values are presented in Figure 10, Figure 11, Figure 12, Figure 13, Figure 14, Figure 15, Figure 16, Figure 17 and Figure 18 and in Table 5, Table 6 and Table 7.

### 3.3. Computational Effort

One of the drawbacks of fractional-, variable-order PID controller is high computational cost, which increases with the duration of the simulation. This is because the calculation of the current value of fractional-, variable-order Grünwald–Letnikov *h*-difference operator, which is used by both difference and summation controller actions, requires all the previous values returned by the controller as shown in Definition 2. It means that FVOPID controller needs an additional buffer, big enough to store mentioned values. Of course, in real-life implementation, the buffer would have to be periodically cleaned. The strategy for efficiently cleaning said buffer of FVOPID controller without significantly disturbing the control is an open topic that requires additional research. When it comes to the number of operations required to calculate the output of FVOPID controller, we will first focus on Definition 1, which describes the oblivion function. In the worst-case scenario, when the order function value is changed in every step and when the given order value did not occur in the past, the controller would have to recalculate all the values of oblivion vector. It means that if we assume that the current step of the simulation is *k*, then *k* calculations of Equation (Equation 2) would have to be executed in this particular step. At the same time, if the order function does not change very often (or does not change at all, like for constant order FOPID controller) the generation of oblivion function vector could be simplified by using previous calculation results (new value can be calculated by single invocation of Equation (Equation 2)). As we can see, in this case, computational cost of FOPID controllers is significantly lower than the one required by FVOPID. With the calculated vector of oblivion function values, backward difference can be evaluated according to Definition 2. For the k−th step of the simulation we would have in this case k+1 multiplication and k+1 summation operations. Additionally, we have to calculate the elements of oblivion function vector to power of the current value of the order function and divide the final result by $h^{v_{k}}$. Additionally, in this step, FOPID controller computational cost could be significantly lower (in comparison to FVOPID) by reusing previously calculated values of oblivion function.

When it comes to the time required by Nelder–Mead optimisation algorithm to find optimal parameters of the controllers, it (as could be assumed) highly depends on the number of searched parameters. Table 8 shows the number of iterations and function evaluations required to find optimal parameters of each designed controller (Nelder–Mead algorithm requires the evaluation of the function at different points in each iteration [25]).

## 4. Discussion

In the first stage of the research, the simulations were carried out with the assumption that the value of control signal produced by the controller is unconstrained. In this case, most of designed FVOPID controllers achieved lower values of minimised errors than constant-order (first and fractional) controllers. Only in one case, when SSTE error was minimised, using FVOPID in which parameters search was based on PID controller parameters, resulted in higher value of error than then one returned by FOPID. In all the cases, fractional-order controllers outperformed Optimal PID which shows how beneficial in this case was the introduction of additional tuning parameters. When it comes to FVOPID controllers, the ones that were tuned with ’FOPID-based‘ algorithm returned lower values of errors than the controllers tuned with ’PID-based‘ algorithm. FVOPID controllers additionally achieved the lowest values of the rise time with the smallest value of the overshoot at the same time. On the other hand, the characteristics returned by PID controllers tend to be smoother (have less oscillations) than corresponding FOPID and FVOPIDs characteristics. It is worth noticing that the selection of minimised error had a great impact on the received results. SSE error minimisation results show short values of rise time and high values of overshoot. At the same time, SST2E error-minimising controllers returned high values of rise time with a very small overshoot.

In the second stage of the research, the controller output values were limited to the values returned by the Initial PID controller (it was assumed that designed controllers’ absolute value of the output signal cannot exceed the absolute maximum value returned by the Initial PID). As we can see, also in this case, the best results when it comes to errors minimisation were achieved by variable order PID. Due to the addition of control signal constraints, FVOPID controller tuned by ’PID-based‘ algorithm gave a little better results than the one tuned with ’FOPID-based‘ algorithm for SSE and SST2E error minimisation. When it comes to the rise time, both FVOPID controllers outperformed the other constant-order (first and fractional) controllers. For overshoot, FVOPID controllers were also able to achieve the lowest values of this criterion, but with the exception of SST2E error minimisation, where FVOPID controller tuned with ’FOPID-based‘ gave slightly worse results than both FOPID and even Optimal PID. In most of the cases, setting restrictions to the controller output signal (as could be predicted) increased obtained error values. On the other hand, characteristics obtained by constrained FVOPID tend to be smoother (smaller oscillations) than the ones which were achieved by the controllers for which output values were not limited. What is also interesting is that in some cases, the order values of FVOPID controllers were negative. It means that given controllers during some phase of the control have two derivative blocks or two integral blocks instead of one derivative and one integral (negative order means that integral action changes to derivative and vice versa).

When it comes to the control signals generated by FVOPID, we can observe a high rate of signal changes in the initial phase of the control, when output value of the process starts to increase and reaches the setpoint. This is the result of changing order values, which are adjusted based on the ratio between the current error and the setpoint. What is worth noticing is that the control signal in some cases reaches negative values, which has to be considered during control process design (not all systems allow negative values of control). Additionally, FVOPID control technique could be implemented in the systems that allow such dynamic control value changes. The potential application could in this case be, e.g., drone engine controllers, where both speed and direction of the propellers rotation could be dynamically changed.

## 5. Conclusions

In the paper, the simulation results of PID; fractional order PID; and fractional-, variable-order PID controllers were presented. Obtained results for the third-order plant with delay shows that for this family of objects/processes usage of FVOPID controller may give better results than the usage of PID/FOPID controllers tuned in a similar way (with Nelder–Mead optimisation). FVOPID control in this case seems to provide additional flexibility and may help to create new control strategies in the future. The main drawbacks of FVOPID controllers are high computational cost and the need to have an additional buffer to store previously returned control values which are used to calculate new controller response. The mentioned disadvantages could be the subject of further research in the future.

## Figures and Tables

**Figure 1 entropy-22-00771-f001:**
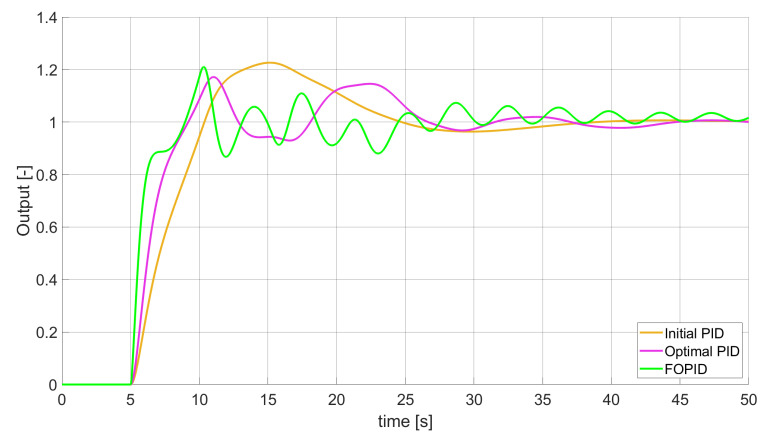
Comparison of the results for SSE error minimisation—constant-order controllers, unconstrained control signal value (Table 2).

**Figure 2 entropy-22-00771-f002:**
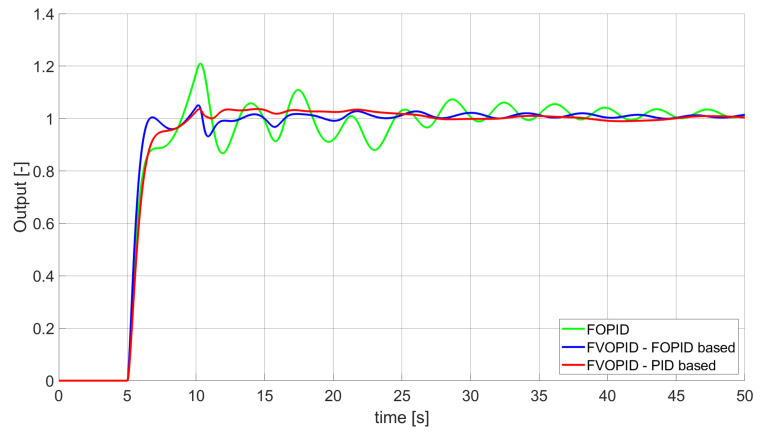
Comparison of the results for SSE error minimisation—fractional-order controllers, unconstrained control signal value (Table 2).

**Figure 3 entropy-22-00771-f003:**
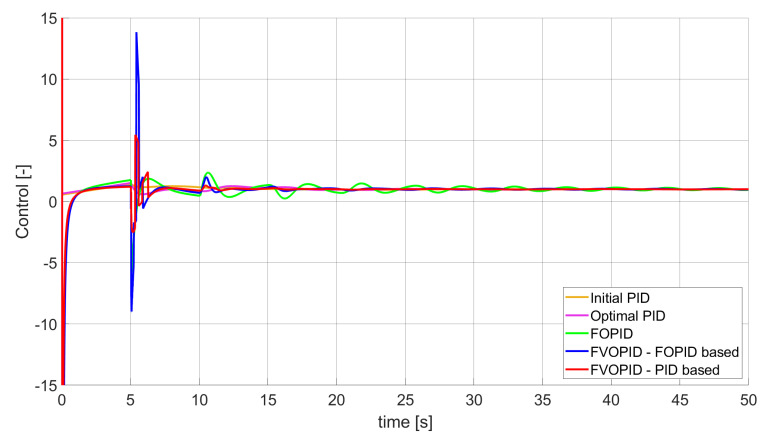
Control signal generated by controllers which minimise SSE error—unconstrained control signal value (Table 2).

**Figure 4 entropy-22-00771-f004:**
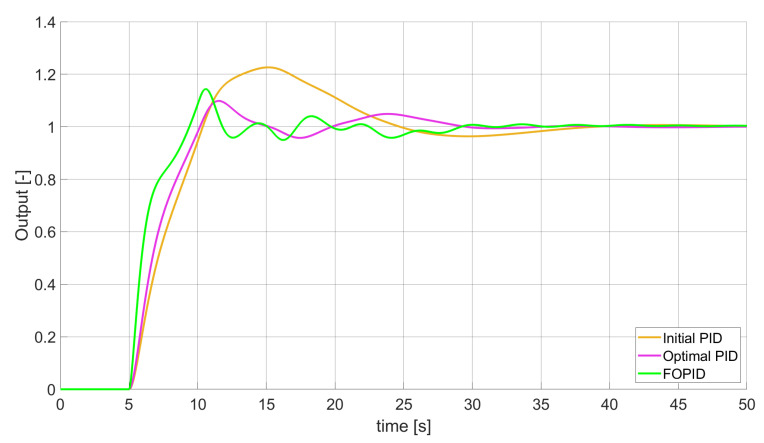
Comparison of the results for SSTE error minimisation—constant-order controllers, unconstrained control signal value (Table 3).

**Figure 5 entropy-22-00771-f005:**
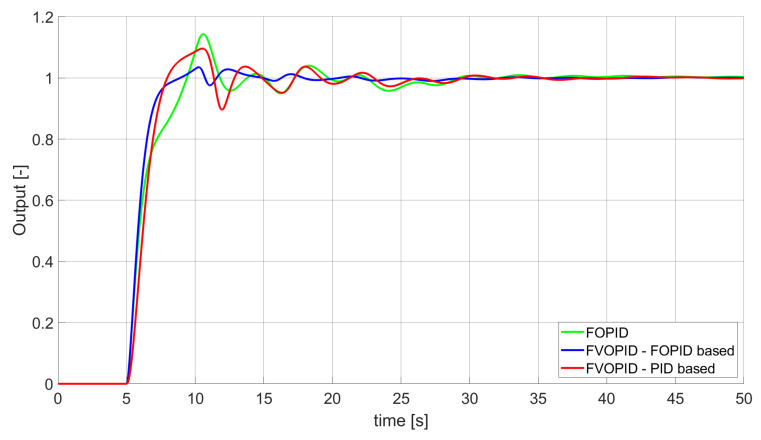
Comparison of the results for SSTE error minimisation—fractional-order controllers, unconstrained control signal value (Table 3).

**Figure 6 entropy-22-00771-f006:**
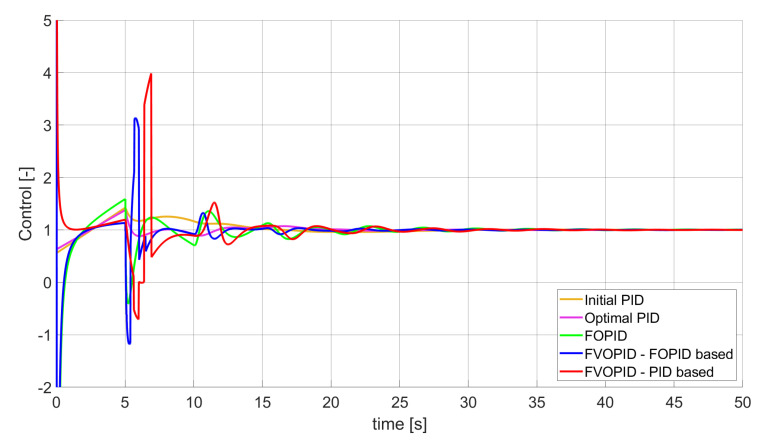
Control signal generated by controllers that minimise SSTE error—unconstrained control signal value (Table 3).

**Figure 7 entropy-22-00771-f007:**
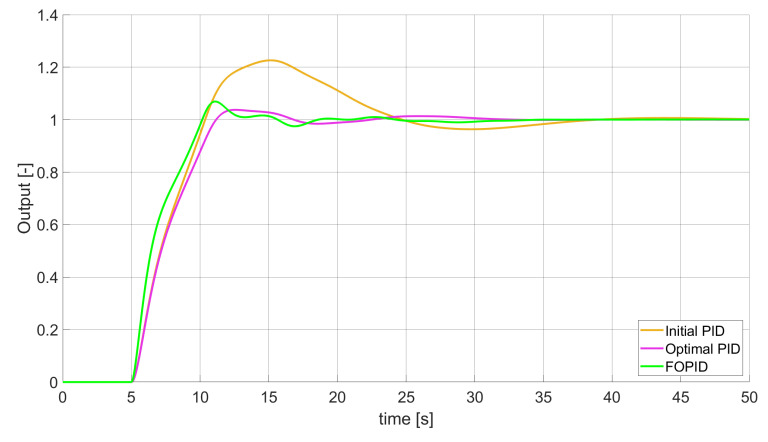
Comparison of the results for SST2E error minimisation—constant-order controllers, unconstrained control signal value (Table 4).

**Figure 8 entropy-22-00771-f008:**
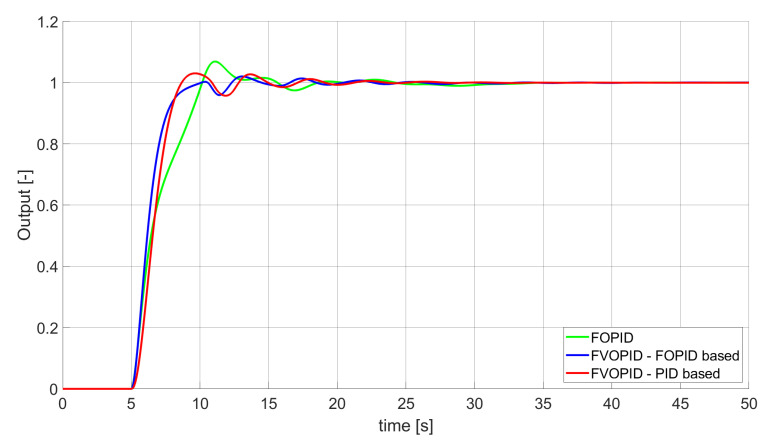
Comparison of the results for SST2E error minimisation—fractional-order controllers, unconstrained control signal value (Table 4).

**Figure 9 entropy-22-00771-f009:**
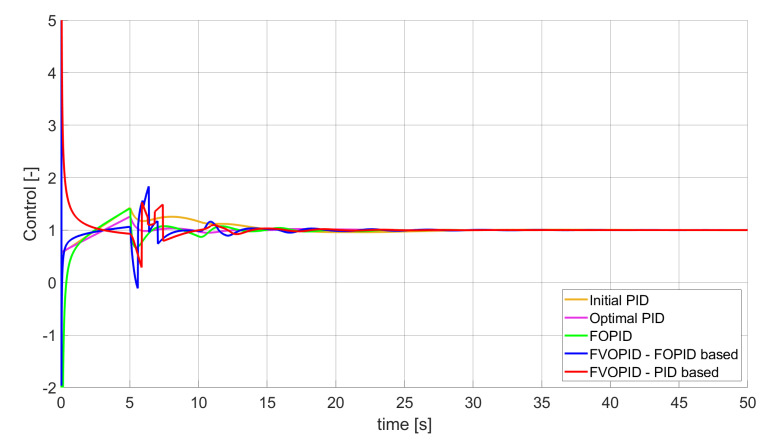
Control signal generated by controllers which minimise SST2E error—unconstrained control signal value (Table 4).

**Figure 10 entropy-22-00771-f010:**
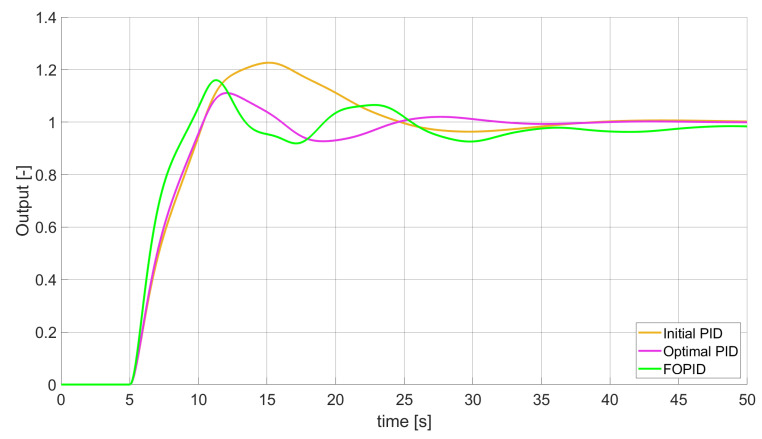
Comparison of the results for SSE error minimisation—constant order controllers, constrained control signal value (Table 5).

**Figure 11 entropy-22-00771-f011:**
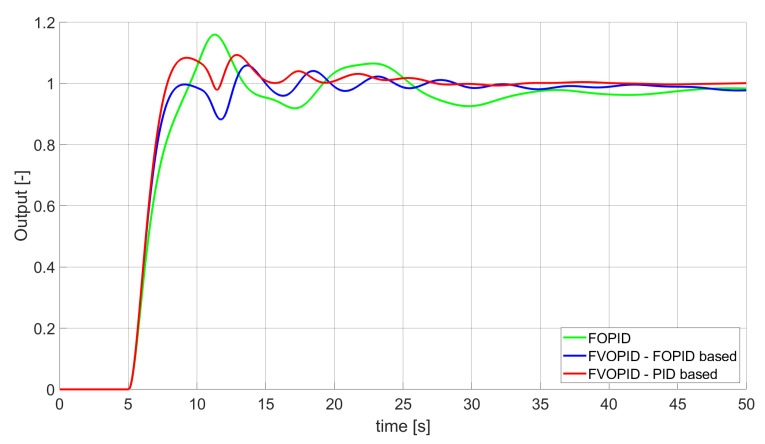
Comparison of the results for SSE error minimisation—fractional order controllers, constrained control signal value (Table 5).

**Figure 12 entropy-22-00771-f012:**
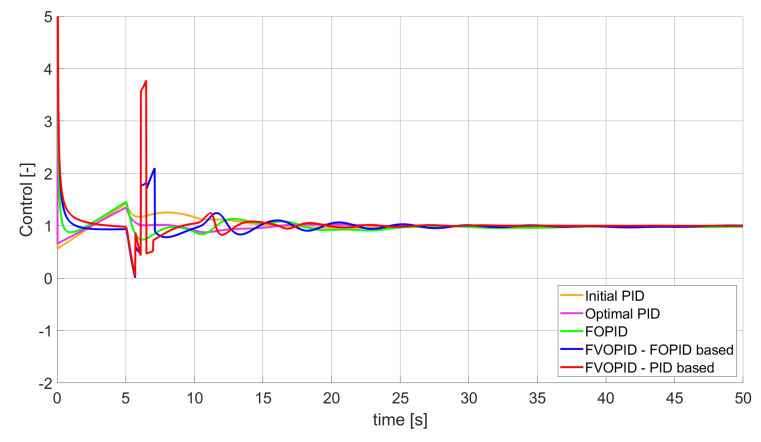
Control signal generated by controllers which minimise SSE error—constrained control signal value (Table 5).

**Figure 13 entropy-22-00771-f013:**
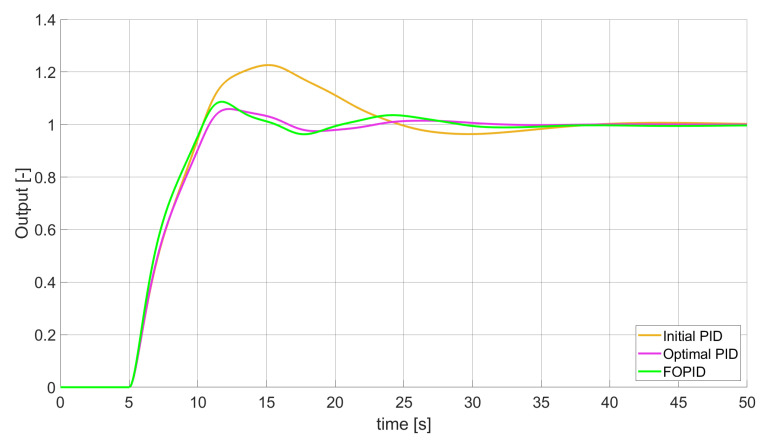
Comparison of the results for SSTE error minimisation—constant-order controllers, constrained control signal value (Table 6).

**Figure 14 entropy-22-00771-f014:**
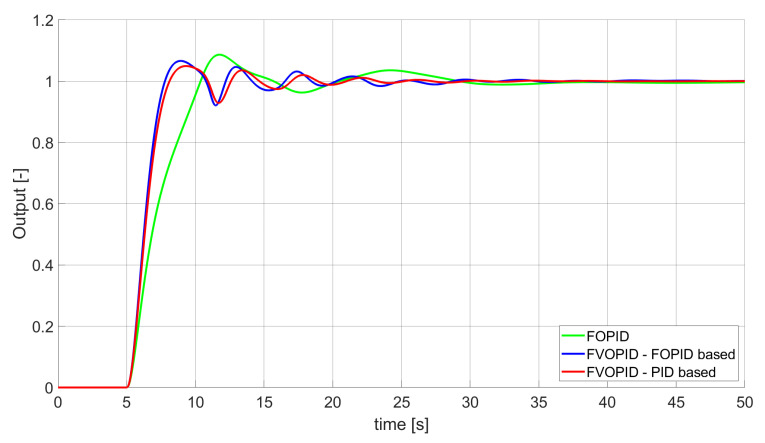
Comparison of the results for SSTE error minimisation—fractional-order controllers, constrained control signal value (Table 6).

**Figure 15 entropy-22-00771-f015:**
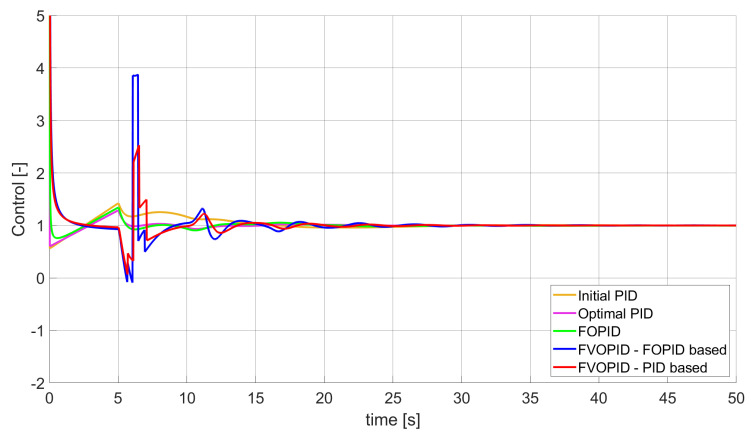
Control signal generated by controllers which minimise SSTE error—constrained control signal value (Table 6).

**Figure 16 entropy-22-00771-f016:**
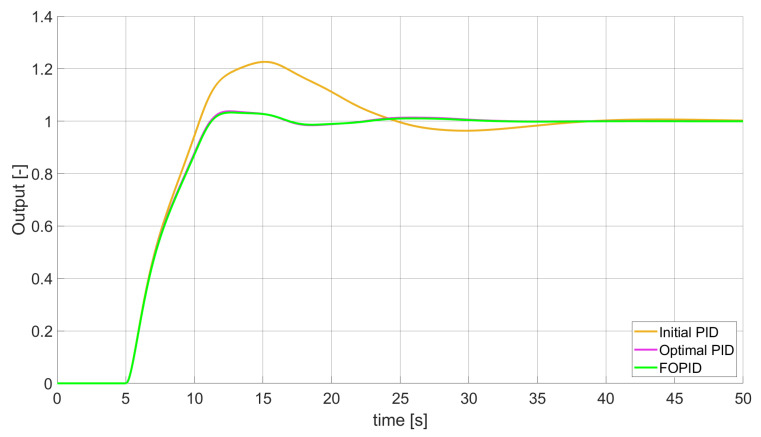
Comparison of the results for SST2E error minimisation—constant-order controllers, constrained control signal value (Table 7).

**Figure 17 entropy-22-00771-f017:**
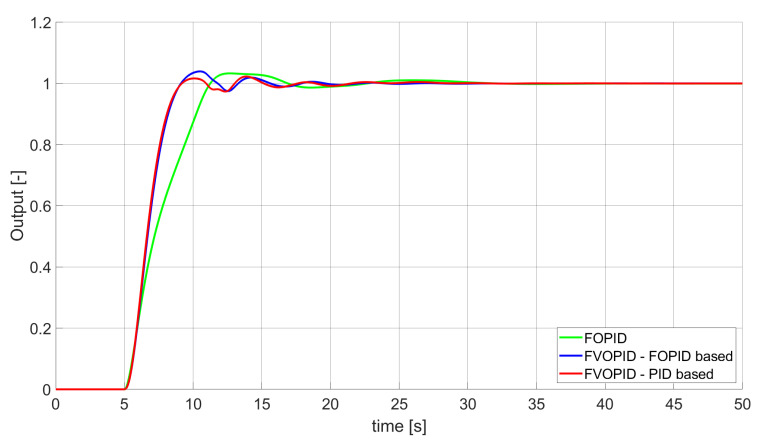
Comparison of the results for SST2E error minimisation—fractional-order controllers, constrained control signal value (Table 7).

**Figure 18 entropy-22-00771-f018:**
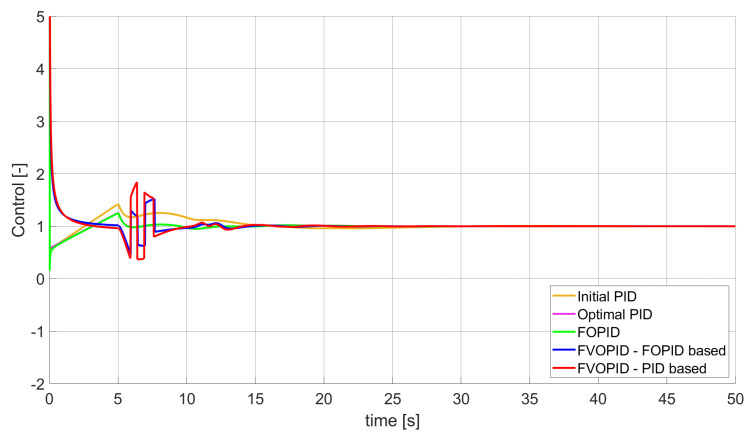
Control signal generated by controllers which minimise SST2E error—constrained control signal value (Table 7).

**Table 1 entropy-22-00771-t001:** Initial Proportional–Integral–Derivative (PID) results.

SSE ^1^	SSTE ^2^	SST2E ^3^	Rise Time	Overshoot	Control Min.	Control Max.
3.438234 $\times 10^{2}$	9.559020 $\times 10^{3}$	1.705381 $\times 10^{6}$	4.1219 s	22.6163	0.5619	48.8435

^1^ Integral squared error ^2^ Integral squared time weighted error ^3^ Integral squared time-squared weighted error.

**Table 2 entropy-22-00771-t002:** SSE minimisation results—unconstrained control signal (Figure 1, Figure 2 and Figure 3).

	Optimal PID	FOPID	FVOPID-FO	FVOPID
$K_{p}$	0.637404	1.142785	1.333838	1.008945
$K_{i}$	0.173603	0.121679	0.159479	0.182504
$K_{d}$	1.777942	2.875904	3.184161	2.814701
$v_{i1}$	1.000000	1.096174	−1.255358	0.205089
$v_{i2}$	1.000000	1.096174	1.329446	0.814011
$v_{i3}$	1.000000	1.096174	1.016607	1.194610
$v_{i4}$	1.000000	1.096174	2.359346	1.738563
$v_{i5}$	1.000000	1.096174	1.039244	1.000000
$v_{d1}$	1.000000	1.498183	1.508917	1.357796
$v_{d2}$	1.000000	1.498183	1.054854	−0.203925
$v_{d3}$	1.000000	1.498183	2.468621	0.998131
$v_{d4}$	1.000000	1.498183	0.974423	1.176751
$v_{d5}$	1.000000	1.498183	1.010596	1.000000
Rise Time	2.7540 s	2.7461 s	0.9507 s	1.4970 s
Overshoot	17.1629	21.0487	5.1426	3.6411
SSTE	3.069558 $\times 10^{2}$	2.798207 $\times 10^{2}$	2.693099 $\times 10^{2}$	2.768806 $\times 10^{2}$
Control Min.	0.5882	−501.8611	−597.4076	−203.0352
Control Max.	89.5359	1.0108 $\times 10^{3}$	1.1887 $\times 10^{3}$	571.6405

**Table 3 entropy-22-00771-t003:** SSTE minimisation results—unconstrained control signal (Figure 4, Figure 5 and Figure 6).

	Optimal PID	FOPID	FVOPID-FO	FVOPID
$K_{p}$	0.626576	0.905323	0.984849	0.602993
$K_{i}$	0.149453	0.144390	0.164533	0.160865
$K_{d}$	1.253809	2.140174	2.416669	2.053849
$v_{i1}$	1.000000	1.024119	0.129042	0.684564
$v_{i2}$	1.000000	1.024119	1.048072	−0.177614
$v_{i3}$	1.000000	1.024119	0.985392	0.842508
$v_{i4}$	1.000000	1.024119	1.009021	1.941464
$v_{i5}$	1.000000	1.024119	1.003853	1.000000
$v_{d1}$	1.000000	1.326989	1.268651	0.889926
$v_{d2}$	1.000000	1.326989	1.947691	0.990233
$v_{d3}$	1.000000	1.326989	2.148687	0.916384
$v_{d4}$	1.000000	1.326989	1.232540	1.689788
$v_{d5}$	1.000000	1.326989	1.083751	1.000000
Rise Time	3.8204 s	3.2553 s	1.6809 s	1.8856 s
Overshoot	9.8411	14.3733	3.5313	9.6220
SSTE	4.917492 $\times 10^{3}$	3.672276 $\times 10^{3}$	3.056599 $\times 10^{3}$	3.779749 $\times 10^{3}$
Control Min.	0.6326	−124.8473	−91.7940	−0.7087
Control Max.	63.3196	385.4887	346.7894	67.3750

**Table 4 entropy-22-00771-t004:** SST2E minimisation results—unconstrained control signal (Figure 7, Figure 8 and Figure 9).

	Optimal PID	FOPID	FVOPID-FO	FVOPID
$K_{p}$	0.576304	0.728516	0.733325	0.708525
$K_{i}$	0.135226	0.142496	0.154479	0.150976
$K_{d}$	0.933056	1.492910	1.929674	1.264259
$v_{i1}$	1.000000	1.005405	0.411674	−0.306919
$v_{i2}$	1.000000	1.005405	0.681375	1.055421
$v_{i3}$	1.000000	1.005405	1.423439	1.199411
$v_{i4}$	1.000000	1.005405	1.005424	1.273247
$v_{i5}$	1.000000	1.005405	1.003021	1.000000
$v_{d1}$	1.000000	1.185370	1.025674	0.680862
$v_{d2}$	1.000000	1.185370	2.117634	0.415573
$v_{d3}$	1.000000	1.185370	1.521393	1.035127
$v_{d4}$	1.000000	1.185370	1.425342	1.179871
$v_{d5}$	1.000000	1.185370	1.062023	1.000000
Rise Time	4.5937 s	3.9967 s	2.2262 s	2.2649 s
Overshoot	3.7552	6.9269	2.0505	3.0619
SST2E	2.140708 $\times 10^{5}$	1.407958 $\times 10^{5}$	8.475202 $\times 10^{4}$	1.071856 $\times 10^{5}$
Control Min.	0.5817	−27.8483	−1.9650	0.2866
Control Max.	47.2320	154.8977	107.4541	19.3490

**Table 5 entropy-22-00771-t005:** SSE minimisation results—constrained control signal (Figure 10, Figure 11 and Figure 12).

	Optimal PID	FOPID	FVOPID-FO	FVOPID
$K_{p}$	0.654698	0.381153	0.417026	0.882641
$K_{i}$	0.138119	0.237605	0.253014	0.172904
$K_{d}$	0.963717	1.655254	1.894754	1.594753
$v_{i1}$	1.000000	0.869949	0.257874	−0.520604
$v_{i2}$	1.000000	0.869949	0.837584	0.526714
$v_{i3}$	1.000000	0.869949	1.335904	2.129147
$v_{i4}$	1.000000	0.869949	1.242142	0.477752
$v_{i5}$	1.000000	0.869949	0.877228	1.000000
$v_{d1}$	1.000000	0.863119	0.823752	0.862873
$v_{d2}$	1.000000	0.863119	0.938978	0.576656
$v_{d3}$	1.000000	0.863119	0.727007	0.576989
$v_{d4}$	1.000000	0.863119	1.134938	1.491228
$v_{d5}$	1.000000	0.863119	0.281767	1.000000
Rise Time	3.9919 s	3.0231 s	2.1032 s	1.8657 s
Overshoot	11.0732	15.9887	5.9138	9.3329
SSE	3.266260 $\times 10^{2}$	3.119663 $\times 10^{2}$	3.003019 $\times 10^{2}$	2.994700 $\times 10^{2}$
Control Min.	0.6602	0.7358	0.0087	0.0346
Control Max.	48.8435	48.8388	48.0527	48.8412

**Table 6 entropy-22-00771-t006:** SSTE minimisation results—constrained control signal (Figure 13, Figure 14 and Figure 15).

	Optimal PID	FOPID	FVOPID-FO	FVOPID
$K_{p}$	0.597129	0.574524	0.639936	0.899289
$K_{i}$	0.137571	0.153988	0.157705	0.157568
$K_{d}$	0.964873	1.153461	1.920653	1.519192
$v_{i1}$	1.000000	0.981612	0.093985	−0.801974
$v_{i2}$	1.000000	0.981612	0.471846	0.651051
$v_{i3}$	1.000000	0.981612	2.161123	1.718251
$v_{i4}$	1.000000	0.981612	0.921158	1.271052
$v_{i5}$	1.000000	0.981612	1.005050	1.000000
$v_{d1}$	1.000000	0.954468	0.823224	0.862127
$v_{d2}$	1.000000	0.954468	0.696274	1.116588
$v_{d3}$	1.000000	0.954468	0.240237	1.398955
$v_{d4}$	1.000000	0.954468	1.454721	1.250374
$v_{d5}$	1.000000	0.954468	1.033150	1.000000
Rise Time	4.3902 s	3.9903 s	1.8090 s	1.9914 s
Overshoot	5.8941	8.6595	6.6260	4.9527
SSTE	5.301965 $\times 10^{3}$	5.058847 $\times 10^{3}$	3.700782 $\times 10^{3}$	3.754522 $\times 10^{3}$
Control Min.	0.6026	0.7589	−0.0899	0.0635
Control Max.	48.8434	48.8430	48.8425	48.8244

**Table 7 entropy-22-00771-t007:** SST2E minimisation results—constrained control signal (Figure 16, Figure 17 and Figure 18).

	Optimal PID	FOPID	FVOPID-FO	FVOPID
$K_{p}$	0.576299	0.574701	0.717493	0.580432
$K_{i}$	0.135232	0.135173	0.147318	0.147948
$K_{d}$	0.933059	0.932648	1.082939	1.271947
$v_{i1}$	1.000000	0.998071	0.113885	0.165591
$v_{i2}$	1.000000	0.998071	1.170532	1.327732
$v_{i3}$	1.000000	0.998071	0.731435	0.638619
$v_{i4}$	1.000000	0.998071	1.206933	1.320971
$v_{i5}$	1.000000	0.998071	1.001298	1.000000
$v_{d1}$	1.000000	1.008807	0.708666	0.652788
$v_{d2}$	1.000000	1.008807	0.856360	1.989044
$v_{d3}$	1.000000	1.008807	1.256436	1.216525
$v_{d4}$	1.000000	1.008807	1.393318	0.268185
$v_{d5}$	1.000000	1.008807	1.027671	1.000000
Rise Time	4.5937 s	4.6411 s	2.5475 s	2.4746 s
Overshoot	3.7552	3.2868	3.9346	2.2612
SST2E	2.140708 $\times 10^{5}$	2.070077 $\times 10^{5}$	1.147133 $\times 10^{5}$	1.119850 $\times 10^{5}$
Control Min.	0.5817	0.1554	0.4744	0.3686
Control Max.	47.2320	48.8432	18.1338	17.0081

**Table 8 entropy-22-00771-t008:** Parameters search Nelder–Mead number of iterations/number of cost function evaluations.

	Opt. PID	FOPID	FVOPID-FO	FVOPID
SSE	63/118	196/334	949/1433	358/645
SSE–constrained	69/130	214/392	380/719	687/1068
SSTE	73/128	156/268	731/1171	286/569
SSTE–constrained	84/150	223/387	932/1447	517/890
SST2E	73/134	153/266	524/904	521/890
SST2E–constrained	69/127	144/255	604/1025	447/773
Average	**72/131**	**181/317**	**687/1117**	**469/809**

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
