# Peer review of "Discrete-Time Fractional, Variable-Order PID Controller for a Plant with Delay"

_entropy, 2020, doi:10.3390/e22070771_

Round 1

Reviewer 1 Report

In my opinion, the paper presents some details. A MAJOR revision is required to make the manuscript worth publishing.

Just to be clear, I believe this is a very good work and contains a novelty. I recommend the publication of the article after the following suggestions

  1. What is the advantage of Fractional derivatives respect to classical derivatives?
  2. What is physical basis of the fractional operators with different memory?
  3. Compare them with those available in the literature, also including discussions on potential applications.
  4. English of whole paper should be checked for gramar.
  5. The figures obtained are not clear, more figures should be obtained with different parameters and with much more detail.
  1. Grammatical mistakes should be corrected in revised manuscript.
  2. Please clarify the numerical experiment section, should contain comments about the computational cost, the conditioning of the problem, the nature of the approximations in solving the system and a comparison in terms of accuracy and computational cost with other methods existing in the literature.
  3. Check the format of the journal and made all the references according to the journal style.

So, I want to read speedly the last version of paper before publishing if it possible for you.

Reviewer 2 Report

This paper presents the simulation results of PID, fractional order PID and fractional-, variable-order PID controllers. FVOPID control seems to provide additional flexibility and may be help to create new control strategies/ approaches in the future. The paper can be further improved by (1) Increasing more references, 14 papers are too little for a journal paper; (2) Practical applications are suggested to consider and increase. (3) It is better to compare other control methods in solving the same problem. (4) Any physical experiments made?

Round 2

Reviewer 1 Report

The paper can be accepted

Reviewer 2 Report

The authors have revised their paper according to my comments, but you should carefully check the paper writing again to avoid any errors. The lines in all figures should be drawn with different line styles, such as dot line, dash line etc, it will be easily identified even if it will be printed in black-white printer.